# BagFlip: A Certified Defense against Data Poisoning

**Yuhao Zhang**
University of Wisconsin-Madison
yuhaoz@cs.wisc.edu

**Aws Albarghouthi**
University of Wisconsin-Madison
aws@cs.wisc.edu

**Loris D'Antoni**
University of Wisconsin-Madison
loris@cs.wisc.edu

## Abstract

Machine learning models are vulnerable to data-poisoning attacks, in which an attacker maliciously modifies the training set to change the prediction of a learned model. In a *trigger-less* attack, the attacker can modify the training set but not the test inputs, while in a *backdoor* attack the attacker can also modify test inputs. Existing model-agnostic defense approaches either cannot handle backdoor attacks or do not provide effective certificates (i.e., a proof of a defense). We present BagFlip, a model-agnostic certified approach that can effectively defend against both trigger-less and backdoor attacks. We evaluate BagFlip on image classification and malware detection datasets. BagFlip is equal to or more effective than the state-of-the-art approaches for trigger-less attacks and more effective than the state-of-the-art approaches for backdoor attacks.

## 1 Introduction

Recent works have shown that machine learning models are vulnerable to data-poisoning attacks, where the attackers maliciously modify the training set to influence the prediction of the victim model as they desire. In a *trigger-less* attack [42, 31, 24, 3, 1, 11], the attacker can modify the training set but not the test inputs, while in a *backdoor* attack [29, 34, 41, 30, 5, 21, 25, 27] the attacker can also modify test inputs. Effective attack approaches have been proposed for various domains such as image recognition [12], sentiment analysis [25], and malware detection [30].

Consider the malware detection setting. An attacker can modify the training data by adding a special signature—e.g., a line of code—to a set of benign programs. The idea is to make the learned model correlate the presence of the signature with benign programs. Then, the attacker can sneak a piece of malware past the model by including the signature, fooling the model into thinking it is a benign program. Indeed, it has been shown that if the model is trained on a dataset with only a few poisoned examples, a backdoor signature can be installed in the learned model [30, 27]. Thus, data-poisoning attacks are of great concern to the safety and security of machine learning models and systems, particularly as training data is gathered from different sources, e.g., via web scraping.

Ideally, a defense against data poisoning should fulfill the following desiderata: (1) Construct **effective certificates** (proofs) of the defense. (2) Defend against **both trigger-less and backdoor attacks**. (3) Be **model-agnostic**. It is quite challenging to fulfill all three desiderata; indeed, existing techniques are forced to make tradeoffs. For instance, empirical approaches [10, 40, 20, 26, 36, 13, 33, 9] cannot construct certificates and are likely to be bypassed by new attack approaches [32, 37, 16]. Some certified approaches [39, 35] provide *ineffective* certificates for both trigger-less and backdoor attacks. Some certified approaches [15, 19, 4, 28, 22, 38] provide effective certificates for trigger-less attacks

but not backdoor attacks. Other certification approaches [14, 7, 23] are restricted to specific learning algorithms, e.g., decision trees.

This paper proposes BagFlip, a *model-agnostic certified* approach that uses *randomized smoothing* [6, 8, 18] to effectively defend against both trigger-less and backdoor attacks (Section 4). BagFlip uses a novel smoothing distribution that combines *bagging* of the training set and noising of the training data and the test input by randomly *flipping* features and labels.

Although both bagging-based and noise-based approaches have been proposed independently in the literature, combining them makes it challenging to compute the *certified radius*, i.e., the amount of poisoning a learning algorithm can withstand without changing its prediction.

To compute the certified radius precisely, we apply the Neyman–Pearson lemma to the sample space of BagFlip's smoothing distribution. This lemma requires partitioning the outcomes in the sample space into subspaces such that the likelihood ratio in each subspace is a constant. However, because our sample space is exponential in the number of features, we cannot naively apply the lemma. To address this problem, we exploit properties of our smoothing distribution to design an efficient algorithm that partitions the sample space into polynomially many subspaces (Section 5). Furthermore, we present a *relaxation* of the Neyman–Pearson lemma that further speeds up the computation (Section 6). Our evaluation against existing approaches shows that BagFlip is comparable to them or more effective for trigger-less attacks and more effective for backdoor attacks (Section 7).

## 2   Related Work

Our *model-agnostic* approach BagFlip uses randomized smoothing to compute *effective certificates* for *both* trigger-less and backdoor attacks. BagFlip focuses on feature-and-label-flipping perturbation functions that modify training examples and test inputs. We discuss how existing approaches differ.

Some model-agnostic certified approaches can defend against both trigger-less and backdoor attacks but cannot construct effective certificates. Wang et al. [35], Weber et al. [39] defended feature-flipping poisoning attacks and $l_2$-norm poisoning attacks, respectively. However, their certificates are practically ineffective due to the curse of dimensionality [17].

Some model-agnostic certified approaches construct effective certificates but cannot defend against backdoor attacks. Jia et al. [15] proposed to defend against *general* trigger-less attacks, i.e., the attackers can add/delete/modify examples in the training dataset, by bootstrap aggregating (bagging). Chen et al. [4] extended bagging by designing other selection strategies, e.g., selecting without replacement and with a fixed probability. Rosenfeld et al. [28] defended against label-flipping attacks and instantiated their framework on linear classifiers. Differential privacy [22] can also provide probabilistic certificates for trigger-less attacks, but it cannot handle backdoor attacks, and its certificates are ineffective. Levine and Feizi [19] proposed a deterministic partition aggregation (DPA) to defend against general trigger-less attacks by partitioning the training dataset using a secret hash function. Wang et al. [38] further improved DPA by introducing a spread stage.

Other certified approaches construct effective certificates but are not model-agnostic. Jia et al. [14] provided deterministic certificates *only* for nearest neighborhood classifiers (kNN/rNN), but for both trigger-less and backdoor attacks. Meyer et al. [23], Drews et al. [7] provided deterministic certificates *only* for decision trees but against general trigger-less attacks.

## 3   Problem Definition

We take a holistic view of training and inference as a single deterministic algorithm $A$. Given a dataset $D = \{(\mathbf{x}_1, y_1), \ldots, (\mathbf{x}_n, y_n)\}$ and a (test) input $\mathbf{x}$, we write $A(D, \mathbf{x})$ to denote the prediction of algorithm $A$ on the input $\mathbf{x}$ after being trained on dataset $D$.

We are interested in certifying that the algorithm will still behave "well" after training on a tampered dataset. Before describing what "well" means, to define our problem we need to assume a *perturbation space*—i.e., what possible changes the attacker could make to the dataset. Given a pair $(\mathbf{x}, y)$, we write $\pi(\mathbf{x}, y)$ to denote the set of perturbed examples that an attacker can transform the example $(\mathbf{x}, y)$ into. Given a dataset $D$ and a *radius* $r \geq 0$, we define the *perturbation space* as the set of

datasets that can be obtained by modifying up to $r$ examples in $D$ using the perturbation $\pi$:

$$S_r^\pi(D) = \left\{ \{(\widetilde{\mathbf{x}}_i, \widetilde{y}_i)\}_i \;\middle|\; \forall i. \; (\widetilde{\mathbf{x}}_i, \widetilde{y}_i) \in \pi(\mathbf{x}_i, y_i), \; \sum_{i=1}^n \mathbb{1}_{(\widetilde{\mathbf{x}}_i, \widetilde{y}_i) \neq (\mathbf{x}_i, y_i)} \leq r \right\}$$

**Threat models.** We consider two attacks: one where an attacker can perturb only the training set (a *trigger-less* attack) and one where the attacker can perturb both the training set and the test input (a *backdoor* attack). We assume a backdoor attack scenario where test input perturbation is the same as training one. If these two perturbation spaces can perturb examples differently, we can always over-approximate them by their union. In the following definitions, we assume that we are given a perturbation space $\pi$, a radius $r \geq 0$, and a benign training dataset $D$ and test input $\mathbf{x}$.

We say that algorithm $A$ is robust to a **trigger-less attack** on the test input $\mathbf{x}$ if $A$ yields the same prediction on the input $\mathbf{x}$ when trained on any perturbed dataset $\widetilde{D}$ and the benign dataset $D$. Formally,

$$\forall \widetilde{D} \in S_r^\pi(D). \; A(\widetilde{D}, \mathbf{x}) = A(D, \mathbf{x}) \tag{1}$$

We say that algorithm $A$ is robust to a **backdoor attack** on the test input $\mathbf{x}$ if the algorithm trained on any perturbed dataset $\widetilde{D}$ produces the prediction $A(D, \mathbf{x})$ on any perturbed input $\widetilde{\mathbf{x}}$. Let $\pi(\mathbf{x}, y)_1$ denote the projection of the perturbation space $\pi(\mathbf{x}, y)$ onto the feature space (the attack can only backdoor features of the test input). Robustness to a backdoor attack is defined as

$$\forall \widetilde{D} \in S_r^\pi(D), \; \widetilde{\mathbf{x}} \in \pi(\mathbf{x}, y)_1. \; A(\widetilde{D}, \widetilde{\mathbf{x}}) = A(D, \mathbf{x}) \tag{2}$$

Given a large enough radius $r$, an attacker can always change enough examples and succeed at breaking robustness for either kind of attack. Therefore, we will focus on computing the maximal radius $r$, for which we can prove that Eq 1 and 2 hold for a given perturbation function $\pi$. We refer to this quantity as the *certified radius*. It is infeasible to prove Eq 1 and 2 by enumerating all possible $\widetilde{D}$ because $S_r^\pi(D)$ can be ridiculously large, e.g., $|S_r^\pi(D)| > 10^{30}$ when $|D| = 1000$ and $r = 10$.

**Defining the perturbation function.** We have not yet specified the perturbation function $\pi$, i.e., how the attacker can modify examples. In this paper, we focus on the following perturbation spaces.

Given a bound $s \geq 0$, a **feature-and-label-flipping perturbation**, $\text{FL}_s$, allows the attacker to modify the values of up to $s$ features and the label in an example $(\mathbf{x}, y)$, where $\mathbf{x} \in [K]^d$ (i.e., $\mathbf{x}$ is a $d$-dimensional feature vector with each dimension having $\{0, 1, \ldots, K\}$ categories). Formally,

$$\text{FL}_s(\mathbf{x}, y) = \{(\widetilde{\mathbf{x}}, y') \mid \|\mathbf{x} - \widetilde{\mathbf{x}}\|_0 + \mathbb{1}_{y \neq y'} \leq s\},$$

There are two special cases of $\text{FL}_s$, a **feature-flipping perturbation** $\text{F}_s$ and a **label-flipping perturbation** $\text{L}$. Given a bound $s \geq 0$, $\text{F}_s$ allows the attacker to modify the values of up to $s$ features in the input $\mathbf{x}$ but not the label. $\text{L}$ only allows the attacker to modify the label of a training example. Note that $\text{L}$ cannot modify the test input's features, so it can only be used in trigger-less attacks.

**Example 3.1.** *If $D$ is a binary-classification image dataset, where each pixel is either black or white, then the perturbation function $\text{F}_1$ assumes the attacker can modify up to one pixel per image.*

The goal of this paper is to design a certifiable algorithm that can defend against trigger-less and backdoor attacks (Section 4) by computing the certified radius (Sections 5 and 6). Given a benign dataset $D$, our algorithm certifies that an attacker can perturb $D$ by some amount (the certified radius) without changing the prediction. Symmetrically, if we suspect that $D$ is poisoned, our algorithm certifies that even if an attacker had not poisoned $D$ by up to the certified radius, the prediction would have been the same. The two views are equivalent, and we use the former in the paper.

## 4 BagFlip: Dual-Measure Randomized Smoothing

Our approach, which we call BagFlip, is a model-agnostic certification technique. Given a learning algorithm $A$, we want to automatically construct a new learning algorithm $\bar{A}$ with certified poisoning-robustness guarantees (Eqs. (1) and (2)). To do so, we adopt and extend the framework of *randomized smoothing*. Initially used for test-time robustness, randomized smoothing robustifies a function $f$ by carefully constructing a *noisy* version $\bar{f}$ and theoretically analyzing the guarantees of $\bar{f}$.

Our approach, BagFlip, constructs a noisy algorithm $\bar{A}$ by randomly perturbing the training set and the test input and invoking $A$ on the result. Formally, if we have a set of output labels $\mathcal{C}$, we define the *smoothed* learning algorithm $\bar{A}$ as follows:

$$\bar{A}(D, \mathbf{x}) \triangleq \underset{y \in \mathcal{C}}{\operatorname{argmax}} \Pr_{\dot{D}, \dot{\mathbf{x}} \sim \mu(D, \mathbf{x})} (A(\dot{D}, \dot{\mathbf{x}}) = y), \tag{3}$$

where $\mu$ is a carefully designed probability distribution—the *smoothing distribution*—over the training set and the test input ($\dot{D}, \dot{\mathbf{x}}$ are a sampled dataset and a test input from the smoothing distribution).

We present two key contributions that enable BagFlip to efficiently certify robustness up to large poisoning radii. First, we define a smoothing distribution $\mu$ that combines *bagging* [15] of the training set $D$, and noising [35] of the training set and the test input $\mathbf{x}$ (done by randomly *flipping* features and labels). This combination, described below, allows us to defend against trigger-less and backdoor attacks. However, this combination also makes it challenging to compute the certified radius due to the combinatorial explosion of the sample space. Our second contribution is a partition strategy for the Neyman–Pearson lemma that results in an efficient certification algorithm (Section 5), as well as a relaxation of the Neyman–Pearson lemma that further speeds up certification (Section 6).

**Smoothing distribution.** We describe the smoothing distribution $\mu$ that defends against feature and label flipping. Given a dataset $D = \{(\mathbf{x}_i, y_i)\}$ and a test input $\mathbf{x}$, sampling from $\mu(D, \mathbf{x})$ generates a random dataset and test input in the following way: (1) Uniformly select $k$ examples from $D$ with replacement and record their indices as $w_1, \ldots, w_k$. (2) Modify each selected example $(\mathbf{x}_{w_i}, y_{w_i})$ and the test input $\mathbf{x}$ to $(\mathbf{x}'_{w_i}, y'_{w_i})$ and $\mathbf{x}'$, respectively, as follows: For each feature, with probability $1 - \rho$, uniformly change its value to one of the other categories in $\{0, \ldots, K\}$. Randomly modify labels $y_{w_i}$ in the same way. In other words, each feature will be *flipped* to another value from the domain with probability $\gamma \triangleq \frac{1-\rho}{K}$, where $\rho \in [0, 1]$ is the parameter controlling the noise level.

For $F_s$ (resp. L), we simply do not modify the labels (resp. features) of the $k$ selected examples.

**Example 4.1.** *Suppose we defend against $F_1$ in a trigger-less setting, then the distribution $\mu$ will not modify labels or the test input. Let $\rho = \frac{4}{5}$, $k = 1$. Following Example 3.1, let the binary-classification image dataset $D = \{(\mathbf{x}_1, y_1), (\mathbf{x}_2, y_2)\}$, where each image contains only one pixel. Then, one possible element of $\mu(D, \mathbf{x})$ can be the pair $(\{(\mathbf{x}'_2, y_2)\}, \mathbf{x})$, where $\mathbf{x}'_2 = \mathbf{x}_2$. The probability of this element is $\frac{1}{2} \times \rho = 0.4$ because we uniformly select one out of two examples and do not flip any feature, that is, the single feature retains its original value.*

## 5    A Precise Approach to Computing the Certified Radius

In this section, we show how to compute the certified radius of the smoothed algorithm $\bar{A}$ given a dataset $D$, a test input $\mathbf{x}$, and a perturbation function $\pi$. We focus on binary classification and provide the multi-class case in Appendix B.

Suppose that we have computed the prediction $y^* = \bar{A}(D, \mathbf{x})$. We want to show how many examples we can perturb in $D$ to obtain any other $\widetilde{D}$ so the prediction remains $y^*$. Specifically, we want to find the largest possible radius $r$ such that

$$\forall \widetilde{D} \in S_r^\pi(D), \ \widetilde{\mathbf{x}} \in \pi(\mathbf{x}, y)_1. \ \Pr_{\dot{D}, \dot{\mathbf{x}} \sim \mu(\widetilde{D}, \widetilde{\mathbf{x}})} (A(\dot{D}, \dot{\mathbf{x}}) = y^*) > 0.5 \tag{4}$$

We first show how to *certify* that Eq 4 holds for a given $r$ and then rely on binary search to compute the largest $r$, i.e., the certified radius.[1] In Section 5.1, we present the Neyman–Pearson lemma to certify Eq 4 as it is a common practice in randomized smoothing. In Section 5.2, we show how to compute the certified radius for the distribution $\mu$ and the perturbation functions $F_s$, $FL_s$, and L.

### 5.1    The Neyman–Pearson Lemma

Hereinafter, we simplify the notation and use $o$ to denote the pair $(\dot{D}, \dot{\mathbf{x}})$ and $A(o)$ to denote the prediction of the algorithm training and evaluating on $o$. We further simplify the distribution $\mu(D, \mathbf{x})$

---

[1]It is difficult to get a closed-form solution of the certified radius (as done in [15, 39]) in our setting because the distribution of BagFlip is complicated. We rely on binary search as it is also done in Wang et al. [35], Lee et al. [18]. Appendix D shows a *loose* closed-form bound on the certified radius by KL-divergence [8].

as $\mu$ and the distribution $\mu(\widetilde{D}, \widetilde{\mathbf{x}})$ as $\widetilde{\mu}$. We define the performance of the smoothed algorithm $\bar{A}$ on dataset $D$, i.e., the probability of predicting $y^*$, as $p^* = \mathrm{Pr}_{o \sim \mu}(A(o) = y^*)$.

The challenge of certifying Eq 4 is that we cannot directly estimate the performance of the smoothed algorithm on the perturbed data, i.e., $\mathrm{Pr}_{o \sim \widetilde{\mu}}(A(o) = y^*)$, because $\widetilde{\mu}$ is universally quantified. To address this problem, we use the Neyman–Pearson lemma to find a lower bound lb for $\mathrm{Pr}_{o \sim \widetilde{\mu}}(A(o) = y^*)$. We do so by constructing a worst-case algorithm $\bar{A}^?$ and distribution $\widetilde{\mu}$. Note that we add the superscript ? to denote a worst-case algorithm. Specifically, we minimize $\mathrm{Pr}_{o \sim \widetilde{\mu}}(A^?(o) = y^*)$ while maintaining the algorithm's performance on $\mu$, i.e., keeping $\mathrm{Pr}_{o \sim \mu}(A^?(o) = y^*) = p^*$. We use $\mathcal{A}$ to denote the set of all possible algorithms. We formalize the computation of the lower bound lb as the following constrained minimization objective:

$$\mathrm{lb} \triangleq \min_{\bar{A}^? \in \mathcal{A}} \mathrm{Pr}_{o \sim \widetilde{\mu}}(A^?(o) = y^*) \quad s.t. \; \mathrm{Pr}_{o \sim \mu}(A^?(o) = y^*) = p^* \text{ and } \widetilde{D} \in S_r^\pi(D), \widetilde{\mathbf{x}} \in \pi(\mathbf{x}, y)_1 \tag{5}$$

It is easy to see that lb is the lower bound of $\mathrm{Pr}_{o \sim \widetilde{\mu}}(A(o) = y^*)$ in Eq. 4 because $\bar{A} \in \mathcal{A}$ and $\bar{A}$ satisfies the minimization constraint. Thus, $\mathrm{lb} > 0.5$ implies the correctness of Eq. 4.

We show how to construct $\bar{A}^?$ and $\widetilde{\mu}$ greedily. For each outcome $o = (\dot{D}, \dot{\mathbf{x}})$ in the sample space $\Omega$—i.e., the set of all possible sampled datasets and test inputs—we define the likelihood ratio of $o$ as $\eta(o) = p_\mu(o)/p_{\widetilde{\mu}}(o)$, where $p_\mu$ and $p_{\widetilde{\mu}}$ are the PMFs of $\mu$ and $\widetilde{\mu}$, respectively.

The key idea is as follows: We partition $\Omega$ into finitely many disjoint subspaces $\mathcal{L}_1, \dots, \mathcal{L}_m$ such that the likelihood ratio in each subspace $\mathcal{L}_i$ is some constant $\eta_i \in [0, \infty]$, i.e., $\forall o \in \mathcal{L}_i.\eta(o) = \eta_i$. We can sort and reorder the subspaces by likelihood ratios such that $\eta_1 \geq \dots \geq \eta_m$. We denote the probability mass of $\mu$ on subspace $\mathcal{L}_i$ as $p_\mu(\mathcal{L}_i)$.

**Example 5.1.** *Suppose* $p_\mu(o_1) = \frac{4}{10}, p_{\widetilde{\mu}}(o_1) = \frac{4}{10}, p_\mu(o_2) = \frac{1}{10}, p_{\widetilde{\mu}}(o_2) = \frac{1}{10}, p_\mu(o_3) = \frac{4}{10}, p_{\widetilde{\mu}}(o_3) = \frac{1}{10},$ $p_\mu(o_4) = \frac{1}{10}, p_{\widetilde{\mu}}(o_4) = \frac{4}{10}$. *We can partition $o_1$ and $o_2$ into one subspace $\mathcal{L}$ because $\eta(o_1) = \eta(o_2) = 1$.*

The construction of the $\bar{A}^?$ that minimizes Eq. 5 is a greedy process, which iteratively assigns $A^?(o) = y^*$ for $\mathcal{L}_1, \mathcal{L}_2, \dots$ until the budget $p^*$ is met. The worst-case $\widetilde{\mu}$ can also be constructed greedily by maximizing the top-most likelihood ratios, and we can prove that the worst-case happens when the difference between $\mu$ and $\widetilde{\mu}$ is maximized, i.e., $\widetilde{D}$ and $\widetilde{\mathbf{x}}$ are maximally perturbed. The following theorem adapts the Neyman–Pearson lemma to our setting.

**Theorem 5.1** (Neyman–Pearson Lemma for $\text{FL}_s, \text{F}_s, \text{L}$). *Let $\widetilde{D}$ and $\widetilde{\mathbf{x}}$ be a maximally perturbed dataset and test input, i.e., $|\widetilde{D} \setminus D| = r$, $\|\widetilde{\mathbf{x}} - \mathbf{x}\|_0 = s$, and $\|\widetilde{\mathbf{x}}_i - \mathbf{x}_i\|_0 + \mathbb{1}_{\widetilde{y}_i \neq y_i} = s$, for each perturbed example $(\widetilde{\mathbf{x}}_i, \widetilde{y}_i)$ in $\widetilde{D}$. Let $i_{\mathrm{lb}} \triangleq \mathrm{argmin}_{i \in [1,m]} \sum_{j=1}^{i} p_\mu(\mathcal{L}_j) \geq p^*$. Then, $\mathrm{lb} = \sum_{j=1}^{i_{\mathrm{lb}}-1} p_{\widetilde{\mu}}(\mathcal{L}_j) + \left( p^* - \sum_{j=1}^{i_{\mathrm{lb}}-1} p_\mu(\mathcal{L}_j) \right) / \eta_{i_{\mathrm{lb}}}$.*

We say that lb is *tight* due to the existence of the minimizer $\bar{A}^?$ and $\widetilde{\mu}$ (see Appendix B).

**Example 5.2.** *Following Example 5.1, suppose $\Omega = \{o_1, \dots, o_4\}$ and we partition it into $\mathcal{L}_1 = \{o_3\}, \mathcal{L}_2 = \{o_1, o_2\}, \mathcal{L}_3 = \{o_4\}$, and $\eta_1 = 4, \eta_2 = 1, \eta_3 = 0.25$. Let $p^* = 0.95$, then we have $i_{\mathrm{lb}} = 3$ and $\mathrm{lb} = 0.1 + 0.5 + (0.95 - 0.4 - 0.5)/0.25 = 0.8$.*

### 5.2 Computing the Certified Radius of BagFlip

Computing the certified radius boils down to computing lb in Eq 5 using Theorem 5.1. To compute lb for BagFlip, we address the following two challenges: 1) The argmin of computing $i_{\mathrm{lb}}$ and the summation of lb in Theorem 5.1 depend on the number of subspaces $\mathcal{L}_j$'s. We design a **partition strategy** that partitions $\Omega$ into a polynomial number of subspaces ($O(k^2d)$). Recall that $k$ is the size of the bag sampled from $D$ and $d$ is the number of features. 2) In Theorem 5.1, lb depends on $p_\mu(\mathcal{L}_j)$, which according to its definition (Eq 8) can be computed in exponential time ($O(kd^{2k})$). We propose an **efficient algorithm** that computes these two quantities in polynomial time ($O(d^3 + k^2d^2)$).

We first show how to address these challenges for $\text{F}_s$ and then show how to handle $\text{FL}_s$ and $\text{L}$.

**Partition strategy.** We partition the large sample space $\Omega$ into disjoint subspaces $\mathcal{L}_{c,t}$ that depend on $c$, the number of perturbed examples in the sampled dataset, and $t$, the number of features the clean data and the perturbed data differ on with respect to the sampled data. Intuitively, $c$ takes care

of the bagging distribution and $t$ takes care of the feature-flipping distribution. Formally,

$$\mathcal{L}_{c,t} = \{(\{(\mathbf{x}'_{w_i}, y_{w_i})\}_i, \mathbf{x}') \mid$$

$$\sum_{i=1}^{k} \mathbb{1}_{\mathbf{x}_{w_i} \neq \widetilde{\mathbf{x}}_{w_i}} = c, \tag{6}$$

$$\underbrace{\left(\sum_{i=1}^{k} \|\mathbf{x}'_{w_i} - \mathbf{x}_{w_i}\|_0 + \|\mathbf{x}' - \mathbf{x}\|_0\right)}_{\Delta} - \underbrace{\left(\sum_{i=1}^{k} \|\mathbf{x}'_{w_i} - \widetilde{\mathbf{x}}_{w_i}\|_0 + \|\mathbf{x}' - \widetilde{\mathbf{x}}\|_0\right)}_{\widetilde{\Delta}} = t\} \tag{7}$$

Eq 6 means that there are $c$ perturbed indices sampled in $o$. $\Delta$ (and $\widetilde{\Delta}$) in Eq 7 counts how many features the sampled and the clean data (the perturbed data) differ on. The number of possible subspaces $\mathcal{L}_{c,t}$, which are disjoint by definition, is $O(k^2 d)$ because $0 \leq c \leq k$ and $|t| \leq (k+1)d$.

The next theorems show how to compute the likelihood ratio of $\mathcal{L}_{c,t}$ and $p_\mu(\mathcal{L}_{c,t})$.

**Theorem 5.2** (Compute $\eta_{c,t}$). $\eta_{c,t} = p_\mu(\mathcal{L}_{c,t})/p_{\widetilde{\mu}}(\mathcal{L}_{c,t}) = (\gamma/\rho)^t$, *where $\gamma$ and $\rho$ are parameters controlling the noise level in BagFlip's smoothing distribution $\mu$.*

**Theorem 5.3** (Compute $p_\mu(\mathcal{L}_{c,t})$).

$$p_\mu(\mathcal{L}_{c,t}) = \Pr_{o \sim \mu} (o \text{ satisfies Eq 6}) \Pr_{o \sim \mu} (o \text{ satisfies Eq 7} \mid o \text{ satisfies Eq 6}),$$

*where* $\Pr_{o \sim \mu}(o \text{ satisfies Eq 6}) = \text{Binom}(c; k, \frac{r}{n})$ *is the PMF of the binomial distribution and*

$$T(c,t) \triangleq \Pr_{o \sim \mu} (o \text{ satisfies Eq 7} \mid o \text{ satisfies Eq 6}) = \sum_{\substack{0 \leq \Delta_i, \widetilde{\Delta}_i \leq d, \forall i \in [0,d] \\ \Delta_0 - \widetilde{\Delta}_0 + \ldots + \Delta_c - \widetilde{\Delta}_c = t}} \prod_{i=0}^{c} L(\Delta_i, \widetilde{\Delta}_i; s, d) \gamma^{\Delta_i} \rho^{d - \Delta_i}, \tag{8}$$

*where $L(\Delta, \widetilde{\Delta}; s, d)$ is the same quantity defined in Lee et al. [18].*

**Remark 5.1.** *We can compute $p_{\widetilde{\mu}}(\mathcal{L}_{c,t})$ as $\eta_{c,t} p_\mu(\mathcal{L}_{c,t})$ by the definition of $\eta_{c,t}$.*

**Efficient algorithm to compute Eq 8.** The following algorithm computes $T(c,t)$ efficiently in time $O(d^3 + k^2 d^2)$: 1) Computing $L(u, v; s, d)$ takes $O(d^3)$ (see Appendix C for details). 2) Computing $T(0, t)$ by the definition in Eq 8 takes $O(kd^2)$. 3) Computing $T(c, t)$ for $c \geq 1$ by the following equation takes $O(k^2 d^2)$.

$$T(c, t) = \sum_{t_1 = \max(-d, t - cd)}^{\min(d, t + cd)} T(c - 1, t - t_1) T(0, t_1) \tag{9}$$

**Theorem 5.4** (Correctness of the Algorithm). $T(c, t)$ *in Eq 9 is the same as the one in Eq 8.*

The above computation still applies to perturbation function $\text{FL}_s$ and L. Intuitively, flipping the label can be seen as flipping another dimension in the input features (Details in Appendix B.1).

**Practical perspective.** For each test input $\mathbf{x}$, we need to estimate $p^*$ for the smoothed algorithm $\bar{A}$ given the benign dataset $D$. We use Monte Carlo sampling to compute $p^*$ by the Clopper-Pearson interval. We also reuse the trained algorithms for each test input in the test set by using *Bonferroni correction*. We memoize the certified radius by enumerating all possible $p^*$ beforehand so that checking Eq 4 can be done in constant time in an online scenario (details in Appendix B.2). We cannot use floating point numbers because some intermediate values are too small to store in the floating point number format. Instead, we use rational numbers which represent the nominator and denominator in large numbers.

## 6 An Efficient Relaxation for Computing a Certified Radius

Theorem 5.1 requires computing the likelihood ratio of a large number of subspaces $\mathcal{L}_1, \ldots, \mathcal{L}_m$. We propose a generalization of the Neyman–Pearson lemma that **underapproximates** the subspaces by a small subset and computes a lower bound $\text{lb}_\delta$ for $\text{lb}$. We then show how to choose a subset of subspaces that yields a tight underapproximation.

**A relaxation of the Neyman–Pearson lemma.** The key idea is to use Theorem 5.1 to examine only a *subset* of the subspaces $\{\mathcal{L}_i\}_i$. Suppose that we partition the subspaces $\{\mathcal{L}_i\}_i$ into two sets $\{\mathcal{L}_i\}_{i \in B}$ and $\{\mathcal{L}_i\}_{i \notin B}$, using a set of indices $B$. We define a new lower bound $\mathrm{lb}_\delta$ by applying Theorem 5.1 to the first group $\{\mathcal{L}_i\}_{i \in B}$ and underapproximating $p^*$ in Theorem 5.1 as $p^* - \sum_{i \notin B} p_\mu(\mathcal{L}_i)$. Intuitively, the underapproximation ensures that any subspace in $\{\mathcal{L}_i\}_{i \notin B}$ will not contribute to the lower bound, even though these subspaces may have contributed to the precise $\mathrm{lb}$ computed by Theorem 5.1.

The next theorem shows that $\mathrm{lb}_\delta$ will be smaller than $\mathrm{lb}$ by an error term $\delta$ that is a function of the partition $B$. The gain is in the number of subspaces we have to consider, which is now $|B|$.

**Theorem 6.1** (Relaxation of the Lemma). *Define* $\mathrm{lb}$ *for the subspaces* $\mathcal{L}_1, \ldots, \mathcal{L}_m$ *as in Theorem 5.1. Define* $\mathrm{lb}_\delta$ *for* $\{\mathcal{L}_i\}_{i \in B}$ *as in Theorem 5.1 by underapproximating* $p^*$ *as* $p^* - \sum_{i \notin B} p_\mu(\mathcal{L}_i)$, *then we have* ***Soundness:*** $\mathrm{lb}_\delta \leq \mathrm{lb}$ *and* $\delta$***-Tightness:*** *Let* $\delta \triangleq \sum_{i \notin B} p_{\widetilde{\mu}}(\mathcal{L}_i)$, *then* $\mathrm{lb}_\delta + \delta \geq \mathrm{lb}$.

**Example 6.1.** *Consider* $\mathcal{L}_1, \mathcal{L}_2$, *and* $\mathcal{L}_3$ *from Example 5.2. If we set* $B = \{1, 2\}$ *and underapproximate* $p^*$ *as* $p^* - p_\mu(\mathcal{L}_3) = 0.85$, *then we have* $\mathrm{lb}_\delta = 0.1 + (0.85 - 0.4)/1 = 0.55$, *which can still certify Eq. 4. However,* $\mathrm{lb}_\delta$ *is not close to the original* $\mathrm{lb} = 0.8$ *because the error* $\delta = 0.4$ *is large in this example. Next, we will show how to choose* $B$ *as small as possible while still keeping* $\delta$ *small.*

**Speeding up radius computation.** The total time for computing $\mathrm{lb}$ consists of two parts, (1) the efficient algorithm (Eq 9) computes $T(c, t)$ in $O(d^3 + k^2 d^2)$, and (2) Theorem 5.1 takes $O(k^2 d)$ for a given $p^*$ because there are $O(k^2 d)$ subspaces. Even though we have made the computation polynomial by the above two techniques in Section 5.2, it can still be slow for large bag sizes $k$, which can easily be in hundreds or thousands.

We will apply Theorem 6.1 to replace the bag size $k$ with a small constant $\kappa$. Observe that $p_\mu(\mathcal{L}_{c,t})$ and $p_{\widetilde{\mu}}(\mathcal{L}_{c,t})$, as defined in Section 5.2, are negligible for large $c$, i.e., the number of perturbed indices in the smoothed dataset. Intuitively, if only a small portion of the training set is perturbed, it is unlikely that we select a large number of perturbed indices in the smoothed dataset. We underapproximate the full subspaces to $\{\mathcal{L}_{c,t}\}_{(c,t) \in B}$ by choosing the set of indices $B = \{(c, t) \mid c \leq \kappa, |t| \leq (c + 1)d\}$, where $\kappa$ is a constant controlling the error $\delta$, which can be computed as $\delta = \sum_{i \notin B} p_{\widetilde{\mu}}(\mathcal{L}_i) = 1 - \sum_{c=0}^{\kappa} \mathrm{Binom}(c; k, \frac{r}{n})$. Theorem 6.1 reduces the number of subspaces to $|B| = O(\kappa^2 d)$. As a result, all the $k$ appearing in previous time complexity can be replaced with $\kappa$.

**Example 6.2.** *Suppose* $k = 150$ *and* $\frac{r}{n} = 0.5\%$, *choosing* $\kappa = 6$ *leads to* $\delta = 1.23 \times 10^{-5}$. *In other words, it speeds up almost 625X for computing* $T(c, t)$ *and for applying Theorem 5.1.*

## 7 Experiments

The implementation of BagFlip is publicly available[2]. In this section, we evaluate BagFlip against trigger-less and backdoor attacks and compare BagFlip with existing work. Note that we apply the relaxation in Section 6 to BagFlip in all experiments and set $\delta = 10^{-4}$.

**Datasets.** We conduct experiments on MNIST, CIFAR10, EMBER [2], and Contagio (`http://co ntagiodump.blogspot.com`). MNIST and CIFAR10 are datasets for image classification. EMBER and Contagio are datasets for malware detection, where data poisoning can lead to disastrous security consequences. To align with existing work, we select subsets of MNIST and CIFAR10 as MNIST-17, MNIST-01, and CIFAR10-02 in some experiments, e.g., MNIST-17 is a subset of MNIST with classes 1 and 7. We discretize all the features when applying BagFlip. We use clean datasets except in the comparison with RAB [39], where we follow their experimental setup and use backdoored datasets generated by BadNets [12]. A detailed description of the datasets is in Appendix E.1.

**Models.** We train neural networks for MNIST, CIFAR10, EMBER and random forests for Contagio. Unless we specifically mention the difference in some experiments, whenever we compare BagFlip to an existing work, we will use the same network structures, hyper-parameters, and data augmentation as the compared work, and we train $N = 1000$ models and set the confidence level as 0.999 for each configuration. All the configurations we used can be found in the implementation.

**Metrics.** For each test input $(\mathbf{x}_i, y_i)$, algorithm $\bar{A}$ will predict a label and the certified radius $r_i$. Assuming that the attacker had poisoned $R\%$ of the examples in the training set, we define **certified accuracy** as the percentage of test inputs that are correctly classified and their certified radii are no

---

[2] `https://github.com/ForeverZyh/defend_framework`

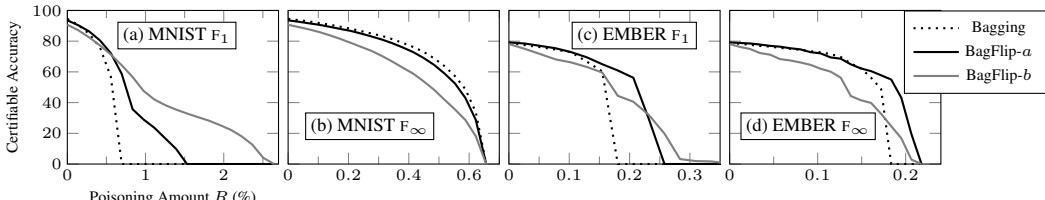

Figure 1: Comparison to Bagging on MNIST and EMBER, showing the certified accuracy at different poisoning amounts $R$. For MNIST: $a = 0.9$ and $b = 0.8$. For EMBER: $a = 0.95$ and $b = 0.9$.

Table 1: Certified accuracy on MNIST and EMBER with perturbations $F_1$ and $F_\infty$. Note that the certified accuracies are the same poisoning amount $R = 0$ because we reuse the trained models.

|  | $R$ | 0 | 0.5 | 1.0 | 1.5 | 2.0 | 2.5 | 0 | 0.13 | 0.27 | 0.40 | 0.53 | 0.83 |
|---|---|---|---|---|---|---|---|---|---|---|---|---|---|
| | | | | $F_1$ | | | | | | $F_\infty$ | | | |
| MNIST | Bagging | **94.54** | 66.84 | 0 | 0 | 0 | 0 | **94.54** | **90.83** | 85.45 | 77.61 | 61.46 | 0 |
| | Bagging-0.9 | 93.58 | 71.11 | 0 | 0 | 0 | 0 | 93.58 | 89.80 | 84.92 | **78.60** | **68.11** | 0 |
| | BagFlip-0.9 | 93.62 | **75.95** | 27.73 | 4.02 | 0 | 0 | 93.62 | 89.45 | 83.62 | 74.19 | 56.65 | 0 |
| | BagFlip-0.8 | 90.72 | 73.94 | **46.20** | **33.39** | **24.23** | **5.07** | 90.72 | 84.11 | 74.50 | 60.56 | 39.21 | 0 |
| | $R$ | 0 | 0.07 | 0.13 | 0.20 | 0.27 | 0.33 | 0 | 0.05 | 0.10 | 0.15 | 0.20 | 0.25 |
| EMBER | Bagging | **82.65** | 75.11 | 66.01 | 0 | 0 | 0 | **82.65** | 76.30 | 72.94 | 61.78 | 0 | 0 |
| | Bagging-0.95 | 79.06 | 75.32 | **70.19** | 14.74 | 0 | 0 | 79.06 | 76.23 | **73.50** | **68.45** | 14.74 | 0 |
| | BagFlip-0.95 | 79.17 | **75.93** | 69.30 | **57.36** | 0 | 0 | 79.17 | **76.83** | 72.41 | 62.04 | **30.36** | 0 |
| | BagFlip-0.9 | 78.18 | 69.40 | 62.11 | 41.21 | **13.89** | **1.70** | 78.18 | 70.79 | 63.16 | 41.24 | 11.64 | 0 |

less than $R$, i.e., $\sum_{i=1}^{m} \mathbb{1}_{\bar{A}(D,\mathbf{x}_i)=y_i \wedge \frac{r_i}{n} \geq R\%}$. We define **normal accuracy** as the percentage of test inputs that are correctly classified, i.e., $\sum_{i=1}^{m} \mathbb{1}_{\bar{A}(D,\mathbf{x}_i)=y_i}$.

### 7.1 Defending Against Trigger-less Attacks

Two model-agnostic certified approaches, Bagging [15] and LabelFlip [28], can defend against trigger-less attacks. BagFlip outperforms LabelFlip (comparison in Appendix E.2). We evaluate BagFlip on the perturbation $F_s$ using MNIST, CIFAR10, EMBER, and Contagio and compare BagFlip to Bagging. We provide comparisons with Bagging on other perturbation spaces in Appendix E.2.

We present BagFlip with different noise levels (different probabilities of $\rho$ when flipping), denoted as BagFlip-$\rho$. When comparing to Bagging, we use the same $k$, the size of sampled datasets, for a fair comparison. Furthermore, we tune the parameter $k$ in Bagging to match the normal accuracy with the BagFlip-$\rho$ setting, and denote this setting as Bagging-$\rho$. Concretely, we set $k = 100, 1000, 300, 30$ for MNIST, CIFAR10, EMBER, and Contagio respectively when comparing to Bagging. We tune $k = 80, 280$ for Bagging-0.9 on MNIST and Bagging-0.95 on EMBER, respectively. And we set $k = 50$ for MNIST when comparing to LabelFlip.

**Comparison to Bagging.** Bagging on a discrete dataset is a special case of BagFlip when $\rho = 1$, i.e., no noise is added to the dataset. (We present the results of bagging on the original dataset (undiscretized) in Appendix E.2). Table 1 and Fig 1 show the results of BagFlip and Bagging on perturbations $F_1$ and $F_\infty$ over MNIST and EMBER. The results of CIFAR10 and Contagio are similar and shown in Appendix E.2. $F_1$ only allows the attacker to modify one feature in each training example, and $F_\infty$ allows the attacker to modify each training example arbitrarily without constraint. For $F_1$ on MNIST, BagFlip-0.9 performs better than Bagging after $R = 0.19$ and BagFlip-0.8 still retains non-zero certified accuracy at $R = 2.5$ while Bagging's certified accuracy drops to zero after $R = 0.66$. We observe similar results on EMBER for $F_1$ in Table 1, except $R = 0.13$ when compared to Bagging-0.95. We argue that it is possible to tune a best combination of $k$ and $\rho$ for BagFlip, like we tune $k$ for Bagging-0.95, and achieve a better result while maintaining similar normal accuracy. However, we do not conduct hyperparameter-tuning for BagFlip because of its computation cost. **BagFlip achieves higher certified accuracy than Bagging when the poisoning amount is large for $F_1$.**

For $F_\infty$ on MNIST, Bagging performs better than BagFlip across all $R$ because the noise added by BagFlip to the training set hurts the accuracy. However, we find that the noise added by BagFlip helps

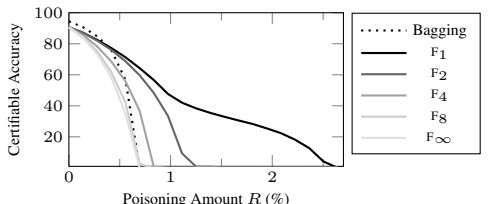 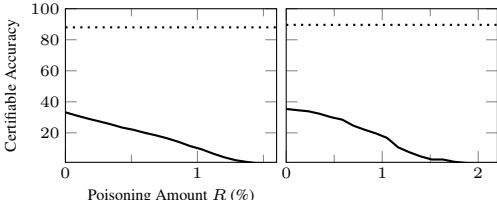

Figure 2: (a) BagFlip-0.8 on MNIST against $F_s$ with different $s$. $s = 8$ almost overlaps with $s = \infty$. (b) BagFlip-0.8 on MNIST and BagFlip-0.9 on Contagio against backdoor attack with $F_1$. Dashed lines show normal accuracy.

it perform better for $F_\infty$ on EMBER. Specifically, BagFlip achieves similar certified accuracy as Bagging at small radii and BagFlip performs better than Bagging after $R = 0.15$. **Bagging achieves higher certified accuracy than BagFlip for $F_\infty$. Except in EMBER, BagFlip achieves higher certified accuracy than Bagging when the amount of poisoning is large.**

We also study the effect of different $s$ in the perturbation function $F_s$. Figure 2(a) shows the result of BagFlip-0.8 on MNIST. BagFlip has the highest certified accuracy for $F_1$. As $s$ increases, the result monotonically converges to the curve of BagFlip-0.8 in Figure 1(b). Bagging neglects the perturbation function and performs the same across all $s$. Bagging performs better than BagFlip-0.8 when $s \geq 8$. Other results for different datasets and different noise levels follow a similar trend (see Appendix E.2). **BagFlip performs best at $F_1$ and monotonically degenerates to $F_\infty$ as $s$ increases.**

## 7.2 Defending Against Backdoor Attacks

Two model-agnostic certified approaches, FeatFlip [35] and RAB [39], can defend against backdoor attacks. We compare BagFlip to FeatFlip on MNIST-17 perturbed using $FL_1$, and compare BagFlip to RAB on poisoned MNIST-01 and CIFAR10-02 (by BadNets [12]) perturbed using $F_1$. We further evaluate BagFlip on the perturbation $F_1$ using MNIST and Contagio, on which existing work is unable to compute a meaningful certified radius.

**Comparison to FeatFlip.** FeatFlip scales poorly compared to BagFlip because its computation of the certified radius is polynomial in the size of the training set. BagFlip's computation is polynomial in the size of the bag instead of the size of the whole training set, and it uses a relaxation of the Neyman–Pearson lemma for further speed up. **BagFlip is more scalable than FeatFlip.**

We directly cite FeatFlip's results from Wang et al. [35] and note that FeatFlip is evaluated on a subset of MNIST-17. As shown in Table 2, BagFlip achieves similar normal accuracy, but much higher certified accuracy across all $R$ (see full results in Appendix E.2) than FeatFlip. **BagFlip significantly outperforms FeatFlip against $FL_1$ on MNIST-17.**

Table 2: Compared to FeatFlip on $FL_1$ and RAB $F_1$ with normal accuracy (Acc.) and certified accuracy at different poisoning amount $R$ (CF@$R$).

|  |  | Acc. | CF@0 | CF@0.5 | CF@1.0 |
|---|---|---|---|---|---|
| MNIST-17 | FeatFlip | 98 | 36 | 0 | 0 |
|  | BagFlip-0.95 | 97 | **81** | **60** | 0 |
|  | BagFlip-0.9 | 97 | 72 | 46 | **4** |

|  |  | Acc. | CF@0 | CF@1.5 | CF@3.0 |
|---|---|---|---|---|---|
| MNIST-01 | RAB | 100.0 | 74.4 | 0 | 0 |
|  | BagFlip-0.8 | 99.6 | **98.4** | **96.5** | 84.6 |
|  | BagFlip-0.7 | 99.5 | 98.0 | 95.8 | **91.9** |

|  |  | Acc. | CF@0 | CF@0.1 | CF@0.2 |
|---|---|---|---|---|---|
| CIFAR10-02 | RAB | 73.3 | 0 | 0 | 0 |
|  | BagFlip-0.8 | 73.1 | 16.8 | 11.0 | 6.8 |
|  | BagFlip-0.7 | 71.7 | **41.0** | **34.6** | **29.2** |

**Comparison to RAB.** RAB assumes a perturbation function that perturbs the input within an $l_2$-norm ball of radius $s$. We compare BagFlip to RAB on $F_1$, where the $l_0$-norm ball (our perturbation function) and $l_2$-norm ball are the same because the feature values are within $[0, 1]$. Since RAB targets single-test-input prediction, we do not use Bonferroni correction for BagFlip as a fair comparison. We directly cite RAB's results from Weber et al. [39]. Table 2 shows that on MNIST-01 and CIFAR10-02, BagFlip achieves similar normal accuracy, but a much higher certified accuracy than RAB for all values of $R$ (detailed figures in Appendix E.2). **BagFlip significantly outperforms RAB against $F_1$ on MNIST-01 and CIFAR10-02.**

**Results on MNIST and Contagio.** Figure 2(b) shows the results of BagFlip on MNIST and Contagio. When fixing $R$, the certified accuracy for the backdoor attack is much smaller than the certified accuracy for the trigger-less attack (Figure 1 and Figure 3 in Appendix E.2) because backdoor attacks are strictly stronger than trigger-less attacks. BagFlip cannot provide effective certificates for backdoor attacks on the more complex datasets CIFAR10 and EMBER, i.e., the certified radius is almost zero. **BagFlip can provide certificates against backdoor attacks on MNIST and Contagio, while BagFlip's certificates are not effective for CIFAR10 and EMBER.**

### 7.3 Computation Cost Analysis

We discuss the computation cost of BagFlip on the MNIST dataset and compare to other baselines.

**Training.** The cost of BagFlip during training is similar to all the baselines because BagFlip only adds noise in the training data. BagFlip and other baselines take about 16 hours on a single GPU to train $N = 1000$ classifiers on the MNIST dataset.

**Inference.** At inference time, BagFlip first evaluates the predictions of $N$ classifiers, and counts how many classifiers have the majority label ($N_1$) and how many have the runner-up label ($N_2$). Then, BagFlip uses a prepared lookup table to query the radius certified by $N_1$ and $N_2$. The inference time for each example contains the evaluation of $N$ classifiers and an O(1) table lookup. Hence, there is no difference between BagFlip and other baselines.

**Preparation.** BagFlip needs to prepare a table of size O($N^2$) to perform efficient lookup at inference time. The time complexity of preparing each table entry is presented in Sections 5 and 6. On the MNIST dataset, BagFlip with the relaxation proposed in Section 6 ($\delta = 10^{-4}$) needs 2 hours to prepare the lookup table on a single core. However, the precise BagFlip proposed in Section 5 needs 85 hours to prepare the lookup table. Bagging also uses a lookup table that can be built in 16 seconds on MNIST (Bagging only needs to do a binary search for each entry). FeatFlip needs approximately 8000 TB of memory to compute its table. Thus, FeatFlip is infeasible to run on the full MNIST dataset. FeatFlip is only evaluated on a subset of the MNIST-17 dataset containing only 100 training examples. RAB does not need to compute the lookup table because it has a closed-form solution for computing the certified radius.

**BagFlip has similar training and inference time compared to other baselines. The relaxation technique in Section 6 is useful to reduce the preparation time from 85 hours to 2 hours.** Even with the relaxation, BagFlip needs more preparation time than Bagging and RAB. We argue that the preparation of BagFlip is feasible because it only takes 12.5% of the time required by training.

## 8 Conclusion, Limitations, and Future Work

We presented BagFlip, a certified probabilistic approach that can effectively defend against both trigger-less and backdoor attacks. We foresee many future improvements to BagFlip. First, BagFlip treats both the data and the underlying machine learning models as closed boxes. Assuming a specific data distribution and training algorithm can further improve the computed certified radius. Second, BagFlip uniformly flips the features and the label, while it is desirable to adjust the noise levels for the label and important features for better normal accuracy according to the distribution of the data. Third, while probabilistic approaches need to retrain thousands of models after a fixed number of predictions, the deterministic approaches can reuse models for every prediction. Thus, it is interesting to develop a deterministic model-agnostic approach that can defend against both trigger-less and backdoor attacks.

## Acknowledgments and Disclosure of Funding

We thank the anonymous reviewers and Anna P. Meyer for their thoughtful and generous comments and Gurindar S. Sohi for giving us access to his GPUs. This work is supported by the CCF-1750965, CCF-1918211, CCF-1652140, CCF-2106707, CCF-1652140, a Microsoft Faculty Fellowship, and gifts and awards from Facebook and Amazon.

