# OpenReview forum: "BagFlip: A Certified Defense Against Data Poisoning"
_NeurIPS.cc/2022/Conference — NeurIPS 2022 Accept_

### Official Review · Reviewer_qMzU · 2022-06-20

**Rating:** 5
**Confidence:** 2
**Soundness:** 2 fair
**Presentation:** 2 fair
**Contribution:** 3 good

**Summary:**

The article proposes a model-agnostic certified method to handle data-poisoning attacks that maliciously modify the training dataset. The article addresses trigger-less and backdoor attacks, and solves them with a noising/smoothing approach on training datasets. The proposed method was evaluated on image and malicious portable executable datasets including MNIST, CIFAR10, and EMBER, and showed robust accuracy in trigger-less and backdoor attacks, respectively.

**Questions:**

1. Section 7.2 describes the differences in the proposed method compared to FeatFlip. However, considering that FeatFlip proposed a randomized smoothing strategy, utilizing additional noise to cope with trigger-less attack seems like a minor modification of FeatFlip. Compared to FeatFlip, what is the technical novelty of the proposed method?
2. While it is true that inductive learning approaches (including machine learning and deep learning) are being explored, many malware detection software is still based on traditional search/traverse algorithms based on signatures. In terms of malware detection, are data poisoning attacks observed in the real world?


**Limitations:**

The proposed method uniformly flips features and labels regardless of data distribution.

**Strengths And Weaknesses:**

(strength)
The article is well-structured, easy-to-follow. The proposed method (BagFlip) is technically sound.
(weakness)
The proposed method was compared only for a limited control group (FeatFlip[35] and RAB[39]). Considering that FeatFlip[35] proposed a randomized smoothing strategy, the technical novelty of the proposed method is weak.

---

> ### Author Response · Authors · 2022-07-29
> **Response to Reviewer qMzU**
>
> We thank reviewer qMzU for their comments and questions.
>
> >**Compared to FeatFlip, what is the technical novelty of the proposed method?**
>
> FeatFlip, RAB, and Bagging all work in the framework of randomized smoothing. Our first contribution is a novel smoothing distribution that combines the distribution used in Bagging and the distribution used in FeatFlip. This combination allows us to defend against trigger-less and backdoor attacks. However, this combination also makes it challenging to compute the certified radius due to the combinatorial explosion of the sample space. Our second contribution is a partitioning strategy for the Neyman--Pearson lemma that results in an efficient certification algorithm and a relaxation of the Neyman--Pearson lemma that further speeds up certification.
>
> Randomized smoothing approaches all suffer from the curse of dimensionality [1]. Compared to FeatFlip, our smoothing distribution adds a bagging strategy, which is crucial to reducing the dimensionality of the sample space, thus, improving the scalability and effectiveness of the certification algorithm. As a result,
> * FeatFlip does not scale to the full MNIST dataset, as FeatFlip needs approximately 8000 TB memory to compute the certified radius.
> * BagFlip significantly outperforms FeatFlip against $FL_1$ on MNIST-17, as shown in Table 2.
>
> [1] Aounon Kumar, Alexander Levine, Tom Goldstein, and Soheil Feizi. Curse of dimensionality on randomized smoothing for certifiable robustness. ICML2020
>
> >**In terms of malware detection, are data poisoning attacks observed in the real world?**
>
> The malware detection approaches can be divided into dynamic and static ones. The static approaches process executable files without running them, extracting the features used for classification directly from the binary and its meta-data. As static approaches, like querying from a maintained malware database, are still used, ML-based approaches have been studied [2,3,4,5,6,7] and deployed in commercial endpoint protection solutions [8,9,10].
> The research community has studied evasion attacks [11,12,13] and data poisoning attacks [14] on malware detection. Evasion attacks have successfully attacked open-source and commercial malware detectors [15,16]. Although we cannot find articles about data poisoning attacks observed in the real world, we believe data poisoning attacks can likely break malware detectors in the future, as evasion attacks have done before.
>
>
> [2] Zheng Leong Chua, Shiqi Shen, Prateek Saxena, and Zhenkai Liang. Neural Nets Can Learn Function Type Signatures From Binaries. In USENIX Security Symposium, 2017
>
> [3] Marek Krcál, Ondrej Švec, Martin Bálek, and Otakar Jašek. Deep Convolutional Malware Classifiers Can Learn from Raw Executables and Labels Only. ICLR 2018
>
> [4] Enrico Mariconti, Lucky Onwuzurike, Panagiotis Andriotis, Emiliano De Cristofaro, Gordon Ross, and Gianluca Stringhini. MaMaDroid: Detecting Android Malware by Building Markov Chains of Behavioral Models. Network and Distributed System Security Symposium 2017
>
> [5] Igor Santos, Felix Brezo, Xabier Ugarte-Pedrero, and Pablo G. Bringas. Opcode sequences as representation of executables for data-mining-based unknown malware detection. Information Sciences, 2013.
>
> [6] Joshua Saxe and Konstantin Berlin. Deep neural network based malware detection using two dimensional binary program features. MALWARE 2015
>
> [7] Andrii Shalaginov, Sergii Banin, Ali Dehghantanha, and Katrin Franke. Machine Learning Aided Static Malware Analysis: A Survey and Tutorial. In Ali Dehghantanha, Mauro Conti, and Tooska Dargahi, Cyber Threat Intelligence, 2018.
>
> [8] https://www.microsoft.com/security/blog/2017/12/11/detonating-a-bad-rabbit-windows-defender-antivirus-and-layered-machine-learning-defenses/
>
> [9] https://www.blackberry.com/us/en/products/cylance-endpoint-security/cylance-is-blackberry-cybersecurity
>
> [10] https://www.fireeye.com/blog/products-and-services/2018/07/malwareguard-fireeye-machine-learning-model-to-detect-and-prevent-malware.html
>
> [11] Battista Biggio, Igino Corona, Davide Maiorca, Blaine Nelson, Nedim Šrndic , Pavel Laskov, Giorgio Giacinto, and Fabio Roli. Evasion Attacks against Machine Learning at Test Time. Advanced Information Systems Engineering, 2013.
>
> [12] Kathrin Grosse, Nicolas Papernot, Praveen Manoharan, Michael Backes, and Patrick McDaniel. Adversarial Examples for Malware Detection. ESORICS 2017
>
> [13] Bojan Kolosnjaji, Ambra Demontis, Battista Biggio, Da- vide Maiorca, Giorgio Giacinto, Claudia Eckert, and Fabio Roli. Adversarial Malware Binaries: Evading Deep Learning for Malware Detection in Executables. EUSIPCO 2018.
>
> [14] Explanation-Guided Backdoor Poisoning Attacks Against Malware Classifiers, Giorgio Severi, Jim Meyer, Scott Coull, Alina Oprea, 30th USENIX Security Symposium, USENIX Security 2021
>
> [15] https://www.elastic.co/blog/machine-learning-static-evasion-competition
>
> [16] https://skylightcyber.com/2019/07/18/cylance-i-kill-you/

---

> > ### Author Response · Authors · 2022-08-09
> > **Second Response to Reviewer qMzU**
> >
> > Dear Reviewer qMzU,
> >
> > As the end of the discussion is approaching, we kindly ask you to consider our responses to your concerns. We are very thankful for your comments and suggestions that helped improve our paper.
> >
> > Best regards,
> > Authors of Paper3023.

---

> > > ### Comment · Reviewer_qMzU · 2022-08-10
> > > **Acknowledging authors' response**
> > >
> > > 1. The authors clarify that the novelty of the method is a novel smoothing distribution combining the distributions of Bagging and FeatFlip. I realized the proposed method's difference through the authors' rebuttals, and I would like to reconsider the contribution (2 fair) of the original review.
> > >
> > > 2. Thanks to the authors for their efforts on Q2 (In terms of malware detection, are data poisoning attacks observed in the real world?) of the original review. I found that ML-based approaches have already been commercialized and, in particular, [15,16] was very helpful in understanding the article's contribution.
> > > Besides, when I self-criticize the original review, I think Q2 of the original review is out of the scope of the article. As the authors claim, there are no cases of data poisoning attacks in the real world yet, but I agree that defenses against them will be helpful in the future.

---

### Official Review · Reviewer_6p2x · 2022-07-11

**Rating:** 6
**Confidence:** 4
**Soundness:** 3 good
**Presentation:** 3 good
**Contribution:** 3 good

**Summary:**

This paper presents BagFlip, a model-agnostic certified approach that utilizes bagging and randomized smoothing to defend against various types of poisoning and backdoor attacks. This paper formulates the theoretical way to compute the certified radius of BagFlip and ways to speed up the computation. This paper evaluates BagFlip in various settings to show its effectiveness.

**Questions:**

See above.

**Limitations:**

See above.

**Strengths And Weaknesses:**

Strengths:

+ Strong attack model.
+ This paper provides a certified guarantee in defending against backdoor attacks.
+ The threat model and the goals are formulated clearly.

Weaknesses:
- Potential runtime problem. Could authors provide empirical evaluations for the running time?

- The explanations of experimental results are unclear. When compared with bagging, the same k is used for both bagging and BagFlip. However, BagFlip adds additional Gaussian noise to the k training examples. Therefore, the noise added by BagFlip is larger than bagging. As a result, the normal accuracy for bagging is higher compared with BagFlip. Could authors use a smaller k for bagging such that bagging and BagFlip have similar normal accuracy? Given the similar normal accuracy, we can compare the robustness of the two methods. This is also applicable in the comparison to FeatFlip and RAB.

---

> ### Author Response · Authors · 2022-07-29
> **Response to Reviewer 6p2x**
>
> We thank reviewer 6p2x for their time and expertise. In the paper, we will add runtime analysis and the experiments suggested in the second question.
>
> >**Comparison with baseline about the runtime.**
>
> The runtime of training (about 16 hours for 1000 classifiers on MNIST on a single GPU) is similar to baselines because BagFlip only adds noise to the bags of the training data.
>
> At inference time, BagFlip first evaluates the predictions of N classifiers, and counts the number of the majority label as $N_1$ and the number of the runner-up label as $N_2$. Then, BagFlip uses a pre-computed lookup table to query the certified radius by $N_1$ and $N_2$. The inference time for each example contains the evaluation of N classifiers and an O(1) table lookup. Hence, there is no difference between BagFlip and other randomized smoothing baselines.
>
> The main computational cost lies in the pre-computation of the lookup table. We show the computational cost of the table for the MNIST dataset as follows (single CPU time):
> * Bagging: 16 seconds
> * BagFlip (with $\delta=1e-4$ in Section 6): 1.9 hours
> * BagFlip (without the technique in Section 6): ~85 hours (We report the estimated runtime because we cannot finish the experiment.)
> * FeatFlip needs approximately 8000 TB of memory to compute the table. Thus, FeatFlip is infeasible to run on the full MNIST dataset. FeatFlip is only evaluated on a subset of the MNIST-17 dataset containing only 100 training examples.
> * RAB does not need to compute the lookup table because it has a closed-form solution for computing the certified radius.
>
> Thus, we draw the following conclusions from the runtime analysis,
> 1. BagFlip has similar training and inference time compared to other baselines.
> 2. With the technique proposed in Section 6, BagFlip needs more pre-computation time than Bagging and RAB. We argue that the pre-computation is feasible because it only takes 12% of the time when compared to training.
> 3. The technique in Section 6 is useful to reduce the pre-computation time (from >85 hours to 1.9 hours).
> 4. BagFlip is more scalable than FeatFlip.
>
> >**Compare with baselines while keeping the normal accuracy the same.**
>
> We thank the reviewer for suggesting this. We tried it now and BagFlip is still better than Bagging (table below). Note that BagFlip is a generalization of Bagging—i.e., Bagging is the same as BagFlip if the noise level alpha=0. In the paper, we will add experiments for MNIST and EMBER when comparing BagFlip with Bagging. We note that this kind of comparison is time-consuming and needs many rounds of tuning to ensure the normal accuracy of the two approaches are the same, so we did not conduct this kind of experiment in the short time given to write the rebuttal for other datasets and baselines.
> For MNIST, we find Bagging with k=80 achieves similar normal accuracy compared to BagFlip-0.9 (k=100) on $F_{1}$:
>
> | R                 | 0         | 0.05      | 0.1       | 0.15     | 0.2 | 0.25 |
> |-------------------|-----------|-----------|-----------|----------|-----|------|
> | Bagging k=80      | 93.58     | 71.11     | 0         | 0        | 0   | 0    |
> | BagFlip-0.9 k=100 | **93.62** | **75.95** | **27.73** | **4.02** | 0   | 0    |
>
> For EMBER, we find Bagging with k=280 achieves similar normal accuracy compared to BagFlip-0.95 (k=300) on $F_{1}$:
>
> | R                  | 0         | 0.07      | 0.13      | 0.20      | 0.27 | 0.33 |
> |--------------------|-----------|-----------|-----------|-----------|------|------|
> | Bagging k=280      | 79.06     | 75.32     | **70.19** | 14.74     | 0    | 0    |
> | BagFlop-0.95 k=300 | **79.17** | **75.93** | 69.30     | **57.36** | 0    | 0    |

---

> > ### Comment · Reviewer_6p2x · 2022-08-07
> > **Thanks for the response**
> >
> > Thanks for the detailed response. I have updated the rating score accordingly.

---

### Official Review · Reviewer_vQcS · 2022-07-11

**Rating:** 6
**Confidence:** 3
**Soundness:** 3 good
**Presentation:** 3 good
**Contribution:** 2 fair

**Summary:**

This paper proposes a certified defense against triggerless and backdoor data poisoning attacks. The authors consider a threat model where the attacker can poison training instances (and the test instance for the backdoor attack) by flipping up to s of the features/label per instance. The defense is based on randomized smoothing, where the smoothing operation involves bagging of the training set and random flipping of features/labels. The certificate provides a (probabilistic) guarantee that up to R instances can be poisoned without changing the model’s prediction on a test instance. The derivation and computation of the certificate (i.e. R) is quite involved, and various relaxations are considered to speed up the computation. Experimental results demonstrate improved certified accuracies compared to bagging alone for triggerless attacks, when the fraction of poisoned instances is high and the number of flips is small. Results for backdoor attacks vary depending on the dataset.

**Questions:**

Is there a reason why the flipping approach by Wang, Cao, Jia & Gong (2020) was not considered as a baseline in the experiments?

In Section 7.2, BagFlip is not found to be effective against backdoor attacks for CIFAR10 and EMBER. Are the authors able to speculate why BagFlip doesn’t work for the higher-dimensional datasets?

I noticed the use of a superscript “?” in line 171 and elsewhere. Is that intentional?

**Limitations:**

These are discussed adequately in Section 8.

**Strengths And Weaknesses:**

*Originality*

The BagFlip method proposed in this paper seems to be a hybrid of methods by Jia, Cao & Gong (2021) and Wang, Cao, Jia & Gong (2020), in the sense that it combines bagging and feature/label flipping. In this sense, it could be viewed as a more incremental paper. However the authors should be credited for the derivation of the certificate, which was quite complex in the bagging/flipping setting. Another benefit of their method is that it encompasses both triggerless and backdoor threat models.

*Quality*

I was generally impressed by the quality of the paper, although I was not able to check the derivations in Sections 5 and 6 carefully. It was good to see a variety of datasets were tested in the experiments, as the conclusions did vary in some cases.

*Significance*

The proposed method (BagFlip) seems to perform similarly to Bagging (Jia, Cao & Gong, 2021) for triggerless attacks in regimes where the certified accuracy remains reasonably high (e.g. above 70%). Given this observation, it’s not clear whether BagFlip should be preferred over Bagging. For instance, if the smoothing operation and certificate for BagFlip is significantly more computationally demanding, then BagFlip may not be preferred. It would be interesting to investigate this further, e.g. by comparing the computational cost of BagFlip versus baselines.

*Clarity*

I enjoyed reading the paper. Sections 5 and 6 were somewhat challenging to read, but that is probably unavoidable due to the complexity of the analysis.

---

> ### Author Response · Authors · 2022-07-29
> **Response to Reviewer vQcS**
>
> We thank reviewer vQcS for their time and expertise. In the paper, we will add runtime analysis and discuss the speculated reasons why BagFlip does not work for higher-dimensional datasets.
> >**Comparison with baseline about the runtime.**
>
> The runtime of training (about 16 hours for 1000 classifiers on MNIST on a single GPU) is similar to baselines because BagFlip only adds noise to the bags of the training data.
>
> At inference time, BagFlip first evaluates the predictions of N classifiers, and counts the number of the majority label as $N_1$ and the number of the runner-up label as $N_2$. Then, BagFlip uses a pre-computed lookup table to query the certified radius by $N_1$ and $N_2$. The inference time for each example contains the evaluation of N classifiers and an O(1) table lookup. Hence, there is no difference between BagFlip and other randomized smoothing baselines.
>
> The main computational cost lies in the pre-computation of the lookup table. We show the computational cost of the table for the MNIST dataset as follows (single CPU time):
> * Bagging: 16 seconds
> * BagFlip (with $\delta=1e-4$ in Section 6): 1.9 hours
> * BagFlip (without the technique in Section 6): ~85 hours (We report the estimated runtime because we cannot finish the experiment.)
> * FeatFlip needs approximately 8000 TB of memory to compute the table. Thus, FeatFlip is infeasible to run on the full MNIST dataset. FeatFlip is only evaluated on a subset of the MNIST-17 dataset containing only 100 training examples.
> * RAB does not need to compute the lookup table because it has a closed-form solution for computing the certified radius.
>
> Thus, we draw the following conclusions from the runtime analysis,
> 1. BagFlip has similar training and inference time compared to other baselines.
> 2. With the technique proposed in Section 6, BagFlip needs more pre-computation time than Bagging and RAB. We argue that the pre-computation is feasible because it only takes 12% of the time when compared to training.
> 3. The technique in Section 6 is useful to reduce the pre-computation time (from >85 hours to 1.9 hours).
> 4. BagFlip is more scalable than FeatFlip.
>
> >**Is there a reason why the flipping approach by Wang, Cao, Jia & Gong (2020) was not considered as a baseline in the experiments?**
>
> We compare with Wang, Cao, Jia & Gong (2020) (we call their approach FeatFlip) as a baseline in section 7.2. BagFlip is more scalable than FeatFlip, and BagFlip significantly outperforms FeatFlip against $FL_{1}$ on MNIST-17.
>
> >**In Section 7.2, BagFlip is not found to be effective against backdoor attacks for CIFAR10 and EMBER. Are the authors able to speculate why BagFlip doesn’t work for the higher-dimensional datasets?**
>
> We speculate there are two possible reasons.
> 1. Randomized smoothing has a non-negligible trade-off between normal accuracy and certified radius for evasion attacks. We still see the gap in the results between higher-dimensional datasets like CIFAR10 in the state-of-the-art randomized smoothing tools [1]. As the backdoor attack is stronger than the evasion attack, we can expect that the gap still exists.
> 2. The smoothing distribution of BagFlip samples a sub-training set much smaller than the original dataset. As the higher-dimensional datasets require more training data, the sub-training set can hurt the normal accuracy of the model.
>
> [1] Double Sampling Randomized Smoothing,Linyi Li, Jiawei Zhang, Tao Xie, Bo Li, ICML2022
>
> >**I noticed the use of a superscript “?” in line 171 and elsewhere. Is that intentional?**
>
> They are intentional. We use the superscript “?” to denote a worst-case algorithm. We will add the missing definition of this notation.

---

> > ### Comment · Reviewer_vQcS · 2022-08-07
> > **Acknowledging authors' response**
> >
> > Thanks for summarizing the computational costs of the different approaches. It would be great to see this in an appendix.
> >
> > > We compare with Wang, Cao, Jia & Gong (2020) (we call their approach FeatFlip) as a baseline in section 7.2
> >
> > Sorry, I somehow missed this. Thanks for clarifying.

---

### Meta-Review · Area_Chair_B5kL · 2022-08-27

**Recommendation:** Accept
**Confidence:** Less certain

**Metareview:**

The paper proposes a new method for certified defense against data poisoning, in both trigger-less and backdoor scenarios. The method augments previous work (Bagging) with random flipping of labels. The latter enables computation of probabilistic certificates, although this results in a huge computational overhead. Various relaxation techniques are proposed to improve the computational burden, bringing the cost to just one order of magnitude "above" the baseline. Experiments show reasonable improvement of defense strength compared to the baselines, although the computational cost remains an issue. Apart from its incremental character and the computational complexity, the method is well executed and theoretically sound.

**Award:**

No

---

### Decision · Program_Chairs · 2022-09-14

Accept